# Sequence Analysis of the Complete Mitochondrial Genome of a Medicinal Plant, *Vitex rotundifolia* Linnaeus f. (Lamiales: Lamiaceae)

**DOI:** 10.3390/genes13050839

**Published:** 2022-05-08

**Authors:** Xiaoli Yu, Zhonggang Duan, Yanjun Wang, Qingxin Zhang, Wei Li

**Affiliations:** 1School of Life Science, Huizhou University, Huizhou 516007, China; duanzhonggang@163.com (Z.D.); yanjun_wang8204@126.com (Y.W.); lazqx@hzu.edu.cn (Q.Z.); 2College of Landscape Architecture and Forestry, Qingdao Agricultural University, Qingdao 266109, China

**Keywords:** *Vitex rotundifolia*, Lamiales, mitochondrial genome, MTPT, phylogenetic analysis

## Abstract

In the present study, we depicted the complete mitochondrial genome of a valuable medicinal plant, *Vitex rotundifolia*. The mitochondrial genome of *V. rotundifolia*, mapped as a circular molecule, spanned 380,980 bp in length and had a GC content of 45.54%. The complete genome contained 38 protein-coding genes, 19 transfer RNAs (tRNAs), and 3 ribosomal RNAs (rRNAs). We found that there were only 38.73% (147.54 kb), 36.28% (138.23 kb), and 52.22% (198.96 kb) of the homologous sequences in the mitochondrial genome of *V. rotundifolia*, as compared with the mitochondrial genomes of *Scutellaria tsinyunensis*, *Boea hygrometrica*, and *Erythranthe lutea*, respectively. A multipartite structure mediated by the homologous recombinations of the three direct repeats was found in the *V. rotundifolia* mitochondrial genome. The phylogenetic tree was built based on 10 species of Lamiales, using the maximum likelihood method. Moreover, this phylogenetic analysis is the first to present the evolutionary relationship of *V. rotundifolia* with the other species in Lamiales, based on the complete mitochondrial genome.

## 1. Introduction

*Vitex rotundifolia* is a land plant which belongs to the Verbenaceae and Lamiales, and it is widely located along the seashores of eastern China, Japan, and Korea [1]. *V. rotundifolia* has long served as a medicinal plant and is described in European pharmacopeias dating back to the year 1618 in relation to its positive effects on injuries and inflammation [2]. In China, *V. rotundifolia* is a traditional Chinese medicine, and its fruit is also listed in Chinese pharmacopeias. To date, most of the previous research has spotlighted the pharmacological characteristics and active ingredients of *V. rotundifolia* [1,3]. There are diverse active compounds, such as flavonoids, phenylpropanoids, and diterpenes, that have been reported in the fruit of *V. rotundifolia* [2]. The molecular mechanism of its active compounds’ synthesis is uncertain. The complete plastid genome of *V. rotundifolia* was newly published last year [4]; however, the mitochondrial genome has not been decoded.

Originating from prokaryotes, and increasing in size by continuing the duplication of the intergenic region and acquiring DNA sequences from the chloroplasts and the nucleus during evolution, the plant mitochondrial genomes range in size from dozens and hundreds of kilobases to a few megabases [5,6,7]. In addition to the different genome sizes, the structure and gene contents are also different in the plant mitochondrial genome. The in vivo structure of plant mitochondrial genomes is enigmatic; however, it has been reported that plant mitochondrial DNAs (mtDNAs) typically interconvert between a ‘master circle’ conformation that consists of all genome sequences and a set of subgenomic circles caused by repeat-mediated recombination [8]. For instance, the complete mitochondrial genome of the monkeyflower *(Mimulus guttatus* DC. line IM62) maps as a master circular molecule which is 525,671 bp in length [9], and that of the cucumber maps as 3 entirely, or largely autonomous, circular molecules that are 1556, 84, and 45 kb in length, respectively [10]. Moreover, the circular map is not an exact depiction of the plant mitochondrial genome structure in vivo, and we use circular maps of plant mitochondrial genomes in genome sequencing publications today, as they are acceptable indicators of genome content and sequencing completeness [9].

Notably, gene contents of the plant mitochondrial genome also display diversity. It has been reported in *Viscum scurruloideum* that an exceptional reduction in gene content is accompanied by its size reduction. Additionally, this reduction especially derives from the loss of respiratory complex I (NADH dehydrogenase), which is also detected in many other angiosperms [7]. In the mitochondrial genome of the *S. conica* and *S. noctiflora*, there are only two or three tRNA genes, which are far fewer than in most angiosperms [11]. Although the genome size, structure, and gene contents of mitochondria have been characterized in plants in the last few years, the number of plant mitochondrial genomes that have been reported is also limited.

Therefore, we sequenced a line of *V. rotundifolia* cultivated at Huizhou University by means of second- and third-generation sequencing technology. Based on these data, we assembled the mitochondrial genome using the GetOrganelle toolkit and NextDenovo software. We characterized the size, structure, and gene content of the mitochondrial genome. We helped to reveal the genetic code of the *V. rotundifolia* mitochondrial genome by showing its organization, repeat elements, the acquisition of sequences from the plastid genome, and the phylogenetic analysis of the ten Lamiales species based on the mitochondrial genome and the genes under selection.

## 2. Materials and Methods

### 2.1. Plant Materials and DNA Extraction

The fresh leaves of a *V. rotundifolia* sample were collected from a line (vr-2101) cultivated at Huizhou University, China (E 113° 51′ 78.06″, N 22° 24′ 19.21″). The fresh leaves were stored in liquid nitrogen, and total genomic DNA were extracted from these samples using 2% CTAB. The assessment of purity and quality of DNA was conducted using 1.0% agarose gel and a Nanodrop spectrophotometer 2000 (Thermo Fisher Scientific, Waltham, MA, USA). The samples of fresh leaves and extracted genomic DNA samples of *V. rotundifolia* were stored in the herbarium of the Huizhou University Life Science College (*V. rotundifolia* number: 202011010526XL).

### 2.2. DNA Sequencing and Mitochondrial Genome Assembly

*V. rotundifolia* total DNA was sequenced using the Nanopore platform (PromethION, Oxford Nanopore Technologies, Oxford, UK) and Illumina Hiseq2500 platform (Illumina, San Diego, CA, USA). Then, we performed de novo assembly using the third-generation sequencing reads and NextDenovo v2.0-β.1 (https://github.com/Nextomics/NextDenovo (accessed on 11 January 2022)). We obtained 2378 contigs, with a total size of 251 Mb. To identify the mitochondrial contigs, the contigs were compared with 9 mitochondrial genomes of the Lamiales species with BLASTn [12]. Meanwhile, the second-generation sequencing reads obtained from the Illumina platform were de novo assembled by GetOrganelle (v1.6.4) [1,13]. Moreover, two circular contigs from Nanopore long-read data were used for resolving the repeat sequences in the mitochondrial graph from the Illumina reads assembled using GetOrganelle. Additionally, the final circular molecules were mapped to the second and third reads with BWA (v0.7.12-r1039) and SAMtools (v0.1.19-44428cd) [14,15] to detect the coverage of the assembled mitochondrial genome.

### 2.3. Mitochondrial Genome Annotation and Genome Alignments of Lamiales

The sequence of the *V. rotundifolia* mitochondrial genome was annotated using the web tool GeSeq (https://chlorobox.mpimp-golm.mpg.de/geseq.html (accessed on 5 February 2022)). Moreover, the mitochondrial genomes of *Ajuga reptans* (NC_023103) and *Rotheca serrata* (NC_049064) served as the reference genomes. Then, Apollo software [16] was used to check the locations of the start and stop codons and the intron/exon boundaries. We used tRNAscan-SE v2.0.7 [17] to identify tRNA genes in the mitochondrial genome. Finally, the circular mitochondrial genome map was drawn using OrganellarGenomeDRAW [18]. We deposited the sequences and annotation results of the two subgenomic circular molecules in GenBank under the accession numbers OK563725.1 and OK563726.1.

To find the mitochondrial synteny blocks of the *V. rotundifolia* compared with the other 9 species in the Lamiales, 9 pairs of mitochondrial genome sequences were aligned using Mauve (version 20150226) with an LCB cutoff of 42 [19]. We used BLASTn with the parameters “-word_size 7 -evalue 1 × 10^−6^” to identify the shared mtDNA of the species pairs.

### 2.4. Analysis of Repeat Sequences and Repeat-Mediated Homologous Recombinations

The web application MISA (https://webblast.ipk-gatersleben.de/misa/ (accessed on 8 February 2022)) was employed to detect the simple sequence repeat (SSR) locations in the two circular molecules of the *V. rotundifolia* mitochondrial genome. Additionally, the minimum numbers of the SSR search were set as 10 for the mononucleotide repeat units, 5 for the dinucleotide repeat units, 4 for the trinucleotide repeat units, and 3 for the tetra-, penta-, and hexanucleotide repeat units. The web application Tandem Repeats Finder (https://tandem.bu.edu/trf/trf.advanced.submit.html (accessed on 12 February 2022)) [20] was engaged to analyze the tandem repeat locations in the two circular molecules of the *V. rotundifolia* mitochondrial genome. The parameters were as follows: 2, 7, and 7 for matches, mismatches, and indels, respectively. The minimum alignment score and maximum period size were set as 50 and 500, respectively. These parameters mostly referred to another report of the mitochondrial genome of *Scutellaria tsinyunensis*, which also belongs to Lamiales, for further comparative analysis [21].

We identified dispersed repeats in the mitochondrial genome using BLASTn with the parameters “-word_size 7 -evalue 1 × 10^−6^”, and a minimum length of 100 bp. Moreover, the sequences of the repeats and 1000 bp of their up and down regions were extracted, and the nanopore long reads were aligned to these sequences by the BWA. We then counted the percentage of reads that supported the alternative conformation (AC) of the repeat pair.

### 2.5. Identification of Plastid Integration

The chloroplast genome of the *V. rotundifolia* deposited in GenBank (NC_050991.1) served as the query. We used the BLASTn module in BLASTn to compare the sequences of the chloroplast and mitochondrial genome of *V. rotundifolia* with the threshold of the e-value set as 1 × 10^−6^ and a minimum match of 100 bp. The positions of mitochondrial plastid sequences (MTPTs) of the circular molecule of the mitochondrial genome were shown by a Circos map drawn using the advanced Circos module in TBtools [22].

### 2.6. Phylogenetic Analysis of the 10 Lamiales Species Based on Shared Mitochondrial Protein Sequences

For revealing the phylogeny of *V. rotundifolia*, we collected 9 mitochondrial genomes of Lamiales from NCBI to construct the phylogenetic tree using *Nicotiana tabacum* and *Solanum lycopersicum* as the outgroup. A total of 26 shared mitochondrial genes, extracted using PhyloSuite (v1.2.1) [23], form all 10 of the mitochondrial genomes. The nucleotide sequences of these genes were aligned with MAFFT (v7.450) [24] implemented in PhyloSuite. The nucleotide sequences were then concatenated and used to construct the phylogenetic trees with the maximum likelihood (ML) method implemented in RAxML (v8.2.4) [25]. The pipeline of RAxML was “raxmlHPC-PTHREADSSSE3 -f a -N 1000 -s vr.phy -n vrtree -m GTRGAMMA -x 551314260 -p 551314260” -o o1,o2 -T 10”. The bootstrap analysis was performed with 1000 replicates. Bayesian inference (BI) analysis was performed using MrBayes (v3.2.7a) [26] on the CIPRES Science Gateway (V. 3.3) [27]. Nucleotide substitution model selection was estimated with jModelTest v2.1.10 [28]. Additionally, the web tool iTOL (https://itol.embl.de (accessed on 15 February 2022)) was used to visualize the phylogenetic trees.

### 2.7. Estimation of Nucleotide Substitution Rates and RNA Editing Predicting

For the assessment of the sequence divergence of the genes in the *V. rotundifolia* mitochondrial genome, pairwise 26 protein-coding gene sequences of the mitochondrial genome of Lamiales were prepared for the calculation of the non-synonymous substitution rate (dN), synonymous substitution rate (dS), and the ratio of dN to dS. The calculation was performed in PAML (v4.9) [29] using the yn00 module, conducting the estimation at verbose = 0, icode = 0, weighting = 0, commonf3×4 = 0, and nDATA = 1. The results of the pairwise dN, dS, and dN/dS values were presented using a boxplot drawn using R-package (ggplot2) [30].

RNA editing events change the nucleotides in RNA molecules of the mitochondrial genomes and increase the conservation of protein sequences across different species [31]. The RNA editing sites of the 35 protein-coding genes of the *V. rotundifolia* mitochondrial genome were predicted using PREP-Mt [31] (http://prep.unl.edu/ (accessed on 6 March 2022)), with a cutoff value of 0.2.

## 3. Results

### 3.1. Organization and Rearrangements of the V. rotundifolia Mitochondrial Genome

To generate the sequencing data for the mitochondrial genome assembly, we used the Illumina and Nanopore sequencing platforms to sequence the DNA sample of *V. rotundifolia*. Overall, 68.68 million reads (Appendix A) and 4.10 Gb of clean data were obtained, respectively. The mitochondrial genome of *V. rotundifolia* was assembled as one circular molecule 380,980 bp in length (Figure 1). The average short and long reads’ coverages of the whole circular molecule were 159.77× and 50.45× (Figure 2), with a GC content of 45.54%. The mitochondrial gene content of the *V. rotundifolia* was 60 unique genes in total, including 38 protein-coding genes and 22 non-protein-coding genes (Table 1 and Table 2). The total length of the protein-coding genes was 34,074 bp, accounting for 7.35% of the total length of the genome. The total length of non-coding proteins was 5129 bp (1.61%), including 30 transfer RNA (tRNA) genes and 3 ribosomal RNA (rRNA) genes (rrn18, rrn5, and rrn26). In addition, we found 8 genes with introns (*ccm*Fc, *nad*5, *rps*3, *rps*10, *nad*1, *nad*7, *nad*2, and *nad*4), containing 22 introns in total.

To identify the genome rearrangements in Lamiales species, we counted the quantity of shared DNA and obtained the Mauve alignment between the *V. rotundifolia* and the other 9 Lamiales species (Figure 3 and Appendix A). It was found that the mitochondrial genome sequence of *V. rotundifolia* showed a low level of synteny and common DNA compared with the other Lamiales species. For example, there were only 38.73% (147.54 kb), 36.28% (138.23 kb), and 52.22% (198.96 kb) of the homologous sequences in the mitochondrial genome of *V. rotundifolia* to the mitochondrial genome of *Scutellaria tsinyunensis*, *Boea hygrometrica*, and *Erythranthe lutea*, respectively.

### 3.2. Repeat Sequences and Repeat-Mediated Homologous Recombinations in the V. rotundifolia Mitochondrial Genomes

We identified the SSRs and tandem repeat sequences of the *V. rotundifolia* mitochondrial genome using MISA and Tandem Repeats Finder. A total of 94 SSRs were detected in the *V. rotundifolia* mitochondrial genome (Appendix A). The results show that the tetranucleotide repeats were predominantly abundant, while hexanucleotide repeats were absent in the *V. rotundifolia* mitochondrial genome. Four tandem repeats were identified in the *V. rotundifolia* mitochondrial genome (Appendix A).

We detected 6 direct repeats which were more than 100 bp in length in the *V. rotundifolia* mitochondrial genome. Then, we mapped the Nanopore long reads to obtain the proportion of read pairs that supported AC of these six repeat pairs (Table 2). The location of these repeat sequences in the circular map of the *V. rotundifolia* mitochondrial genome is shown in the Circos map (Appendix A). For the 2 longer repeats, 5616 and 4148 bp in size, nearly the same number of read pairs supported either the “master circle” conformation or the AC (2 subgenomic circles) mediated by the homologous recombination. Moreover, for the repeat that was 283 bp in size, the number of reads that supported the AC was more than the reads that supported the “master circle” conformation. These results imply a multipartite structure of the *V. rotundifolia* mitochondrial genome, consisting of a “master circle” and two subgenomic circles (Figure 4). These alternate circles represent subgenomes found as alternate, shorter versions of the MT DNA within our assembly of the *V. rotundifolia* mitochondrial genome. The other 3 repeats had a low level (<4%) of recombinants, suggesting that the “master circle” conformation also exists at relatively high substoichiometric levels. It should be taken noted that this model is not the sole representation of the proposed evolution of these repeats.

### 3.3. Identification of Mitochondrial Plastid Sequences (MTPT)

For the identification of plastid-derived fragments in the *V. rotundifolia* mitochondrial genome, we used the plastid genome of *V. rotundifolia*, which served as a reference genome (NC_050991.1). The *V. rotundifolia* mitochondrial genome contained 26 transferred chloroplast fragments. The total length of the MTPT fragments was 8997 bp, accounting for 5.82% of the plastid genome and 2.36% of the mitochondrial genome (Appendix A). The size of MTPT fragments in the circular molecules ranged from 115 to 887 bp.

### 3.4. Phylogenetic Analysis

The phylogenetic analysis was performed based on the 26 common mitochondrial genes of 10 Lamiales species (Appendix A). The model GTR + I+G was selected for ML analyses with 1000 bootstrap (BS) replicates to calculate the BS values of the topology. As a result, most nodes of the tree had support values of ML bootstraps that were more than 90, and BI posterior probabilities that were 1. Phylogenetically, *V. rotundifolia* was close to *Scutellaria tsinyunensis*, *Ajuga reptans*, and *Rotheca serrata*. The five species of Lamiaceae were clustered in a clade in this tree (Figure 5).

### 3.5. Identification of Genes under Selection and RNA Editing Events

To comprehensively analyze the nucleotide substitution rates in the mitochondrial genome of *V. rotundifolia*, we evaluated the pairwise non-synonymous substitution rate (dN), synonymous substitution rate (dS), and ratio of dN to dS of the 26 common mitochondrial genes of the 9 Lamiales species with the yn00 module of PAML (v4.9). The dN/dS values of *ccm*B, *ccm*Fc, and *mtt*B were over 1.0 in most of the Lamiales species, which represented the possible positive selection of these genes (Figure 6 and Appendix A). However, a low dN/dS value was found in *atp*1, *atp*9, *cob*, *cox*1, *cox*2, *cox*3, *nad*1, *nad*4L, *nad*5, *nad*6, and *rps*12, which indicated a possible negative selection of these genes.

We identified 439 C-to-U RNA editing sites in the 35 protein-coding genes of the *V. rotundifolia* mitochondrial genome, and *nad*4 and *ccm*B presented the most editing sites (Figure 7A). Moreover, there were 14 types of amino acid transformation at these RNA editing sites (Figure 7B).

## 4. Discussion

In this study, we sequenced and assembled the mitochondrial genome of a medicinal plant, *V. rotundifolia*. The assembled mitochondrial genome sequence of *V. rotundifolia* was 380,980 bp in length (Figure 1), which was close to the length of 354,073 bp of another plant of the Lamiales family, *Scutellaria tsinyunensis*. A total of 34 protein-coding genes were identified in the mitochondrial genome of *V. rotundifolia*, 2 genes more than *S. tsinyunensis*. A total of 19 tRNA genes, 5 genes fewer than *S. tsinyunensis*, and 3 rRNA genes were also identified in the mitochondrial genome (Table 2).

The genome structure of plant mitochondrial genomes has always been mapped as a single master circle and a collection of subgenomic circles because of the large, direct, repeat recombination [8,10]. Although these circular maps do not present the exact structure of the in vivo structure of the plant mitochondrion, they are reported and accepted as indicators of genome content and sequencing completeness [9]. For example, the mitochondrial genome of *S. tsinyunensis* was reported to have 2 conformations mediated by the recombination of 1 direct repeat 175 bp in length. Similar to *S. tsinyunensis*, we also found that the mitochondrial genome of *V. rotundifolia* can be mapped as a master circle or two subgenomic circles arising from a large direct repeat. For the plant mitochondrial genome alignments, it was reported that angiosperm mtDNAs diverge rapidly at the structural level and show loss of synteny and shared DNA, even when compared with closely related species [32]. The research also demonstrated that only 51% (205 kb) of the 401 kb mitogenome of *Vigna radiata* is homologous to the *Glycine max* mitogenome [33]. In our study, the *V. rotundifolia* mitochondrial genome also only shared 26.12% (99.52 kb) to 52.22% (198.96 kb) homologous sequences with the other 9 species of Lamiales.

Our data showed that tetranucleotide repeats were the most abundant and that hexanucleotide repeats were absent in the *V. rotundifolia* mitochondrial genome. These results are consistent with the report on *S. tsinyunensis*, which is also a species of Lamiales. The most abundant dispersed repeats were forward repeats and palindromic repeats, while reverse repeats were not found in the *V. rotundifolia* mitochondrial genome. The results were also reported in the *S. tsinyunensis* mitochondrial genome and the nine mitochondrial genomes of Lamiales in the literature [21].

The topologies of the phylogenetic trees constructed in our study showed congruence compared with the trees constructed by the sequences of the nine mitochondrial genomes and chloroplast genomes of the Lamiales in the report of *S. tsinyunensis*. Additionally, congruence with the tree constructed by the complete plastome sequences from the families of Lamiaceae, Mazaceae, and Phrymaceae was also presented [4]. In our study, the dN/dS value of *ccm*B, *ccm*Fc, and *mtt*B was over 1.0 in most of the Lamiales species, and a low dN/dS value was found in *atp*1, *atp*9, *cob*, *cox*1, *cox*2, *cox*3, *nad*1, *nad*4L, *nad*5, *nad*6, and *rps*12. These results were also shown in the report on Lamiales [21], thereby enabling us to predict the variation and evolution of these genes. For RNA editing sites, 487 C-to-U RNA editing sites were detected in the PCGs of *S. mukorossi* mitogenome, of which the number detected in our study was close. Similarly, the most editing sites were identified in *ccm*B and *nad*4, and the conversion types of the most editing sites were from proline to leucine and from serine to leucine of the *S. mukorossi* mitogenome [34].

## 5. Conclusions

Currently, the research focusing on the mitochondrial genomes of plants, especially medicinal plants, is still limited and provides little information about the evolution and species authentication based on the mitochondrial genomes. Our novel mitochondrial genome will serve as a tool for us to unfold the phylogeny and species identification of *V. rotundifolia*.

## Figures and Tables

**Figure 1 genes-13-00839-f001:**
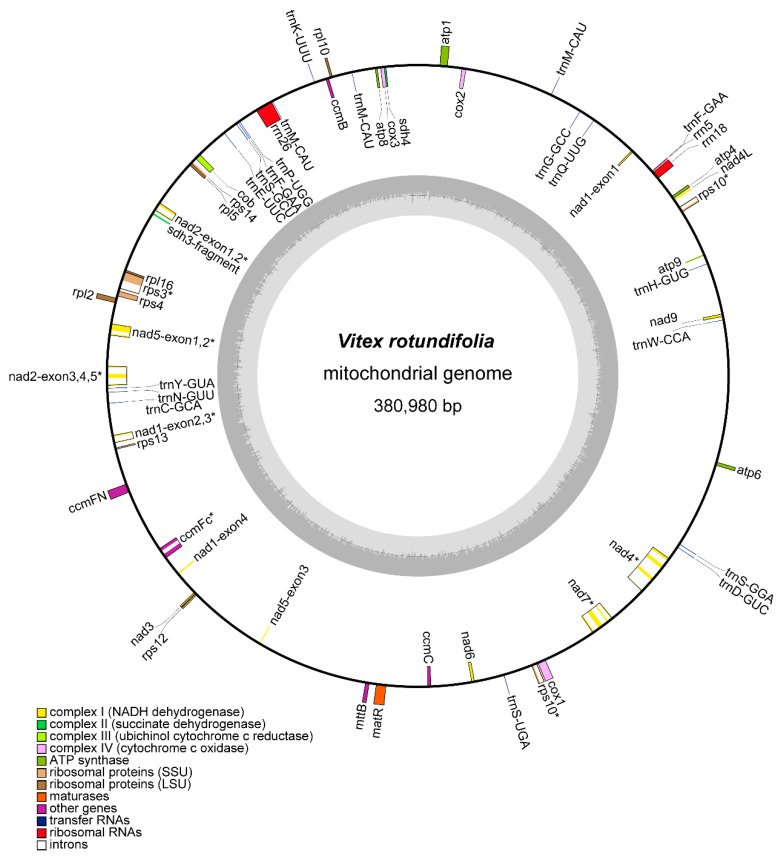
Schematic of the mitochondrial genome of *V. rotundifolia*. Genes on the inside are on the negative strand, and genes on the outside are on the positive strand. The inner grey circle shows the GC contents of the mitochondrial genome. The circle inside the GC content graph shows the 50% threshold. The colors of genes show different functional categories, and details are shown in the legend. Genes with introns were marked with (*).

**Figure 2 genes-13-00839-f002:**
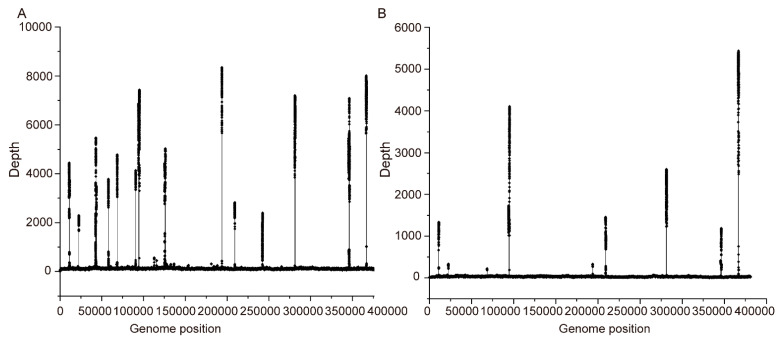
Depth and coverage of the assembled mitogenome using Illumina and Nanopore long reads. Panels (**A**,**B**) represent the depth and coverage of the assembled mitogenome using Illumina and Nanopore long reads, respectively. The x axis shows the genomic positions, and the y axis shows the depth of mapped long reads.

**Figure 3 genes-13-00839-f003:**
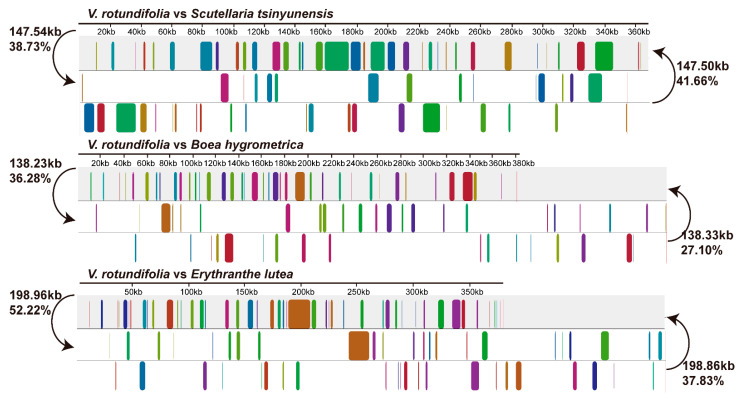
Synteny blocks and shared mtDNA of the *V. rotundifolia* and three other species of the Lamiales based on Mauve alignments. Left arrows show the shared amount of mtDNA, in kb and percentage, of the *V. rotundifolia* and one other species, and right arrows show the reciprocal values.

**Figure 4 genes-13-00839-f004:**
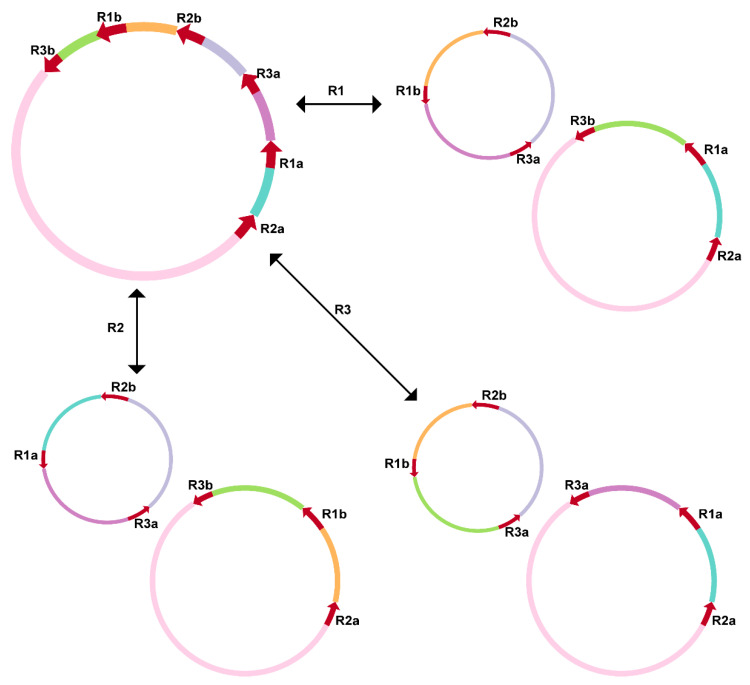
Inferred multipartite structure mediated by the recombination of the three repeats in the *V. rotundifolia* mitochondrial genome. “R1-3a/b”: the pairs of repeat sequences R1–R3 as shown in Table 2. The regions corresponding to different fragments separated by the three repeats are shown in different colors. Note: the circles in the graph resulted from the genome assembly, and they do not necessarily suggest the in vivo state of the *V. rotundifolia* mitochondrial genome.

**Figure 5 genes-13-00839-f005:**
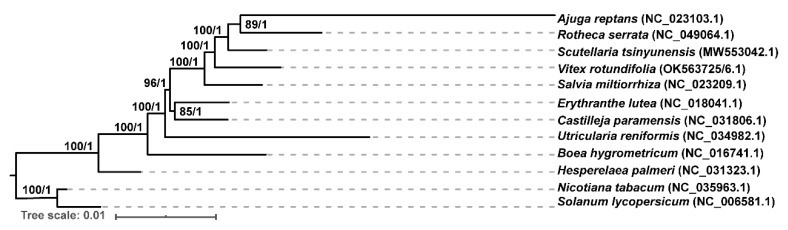
Molecular phylogenetic analysis of mitochondrial genomes of Lamiales. The tree was constructed using shared protein coding gene sequences of the mitochondrial genomes of 10 species using the maximum likelihood and Bayesian inference (BI) method. The bootstrap value was obtained by 1000 replicates. The topology of the tree is indicated with ML bootstrap values and BI posterior probabilities. *Nicotiana tabacum* and *Solanum lycopersicum* were used as outgroups.

**Figure 6 genes-13-00839-f006:**
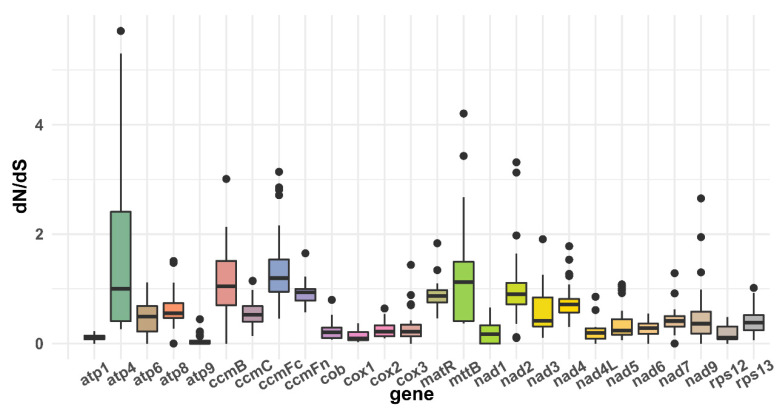
Boxplots of the pairwise dN/dS ratios among every shared mitochondrial gene of the 10 Lamiales species.

**Figure 7 genes-13-00839-f007:**
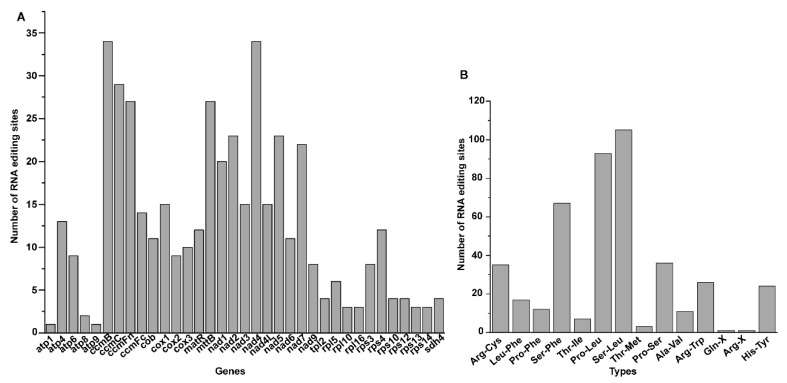
RNA editing sites in the sequences of protein coding genes of *V. rotundifolia*’s mitochondrial genome: (**A**) Number of RNA editing sites of 35 protein coding genes, and (**B**) Number of RNA editing sites leading to different amino acid transformations.

**Table 1 genes-13-00839-t001:** Genes predicted in the mitochondrial genome of *V. rotundifolia*.

Group of Genes	CM1	CM2
Complex I	*nad*1 *, *nad*2 *, *nad*3, *nad*4L, *nad*5, *nad*6	*nad*4 ^#^, *nad*7 ^#^, *nad*9
Complex II	-	-
Complex III	*cob*	-
Complex IV	*cox*2, *cox*3	*cox*1
Complex V	*atp*1, *atp*4, *atp*8	*atp*6, *atp*9
Cytochrome-c biogenesis	*ccm*B, *ccm*C, *ccm*FC ^#^, *ccm*FN	-
SecY-independent transport	*mtt*B	-
Ribosomal protein small subunit	*rps*3 ^#^, *rps*4, *rps*10 ^#^, *rps*12, *rps*13, *rps*14	*rps*10 ^#^
Ribosomal protein large subunit	*rpl*2, *rpl*5, *rpl*10, *rpl*16	-
Intron maturase	*mat*R	-
Ribosomal RNAs	*rrn*5, *rrn*18, *rrn*26	-
Transfer RNA	*trn*C-GCA, *trn*E-UUC, *trn*F-GAA, *trn*G-GCC, *trn*K-UUU, *trn*M-CAU, *trn*M-CAU, *trn*M-CAU, *trn*N-GUU, *trn*P-UGG, *trn*S-GCU, *trn*S-UGA, *trn*Y-GUA, *trn*K-UUU, *trn*Q-UUG	*trn*H-GUG, *trn*W-CCA, *trn*S-GGA, *trn*D-GUC
Succinate dehydrogenase	*sdh*3 *, *sdh*4	-

“^#^”represents genes with more than two exons. “*” represents pseudogenes. -, absent.

**Table 2 genes-13-00839-t002:** Nanopore long reads support the recombination related to six repeat pairs in the mitochondrial genome of *V. rotundifolia*.

Repeat	Length (bp)	Direction	Position	Reads Support Master Circle Conformation	Reads Support Master Circle Conformation
R1	5616	+	1–5616	40	48
		+	265,249–270,864	45.45%	54.55%
R2	4148	+	30,846–34,992	66	64
		+	305,624–309,771	50.77%	49.23%
R3	283	+	264,966–265,248	10	16
		+	380,698–380,980	38.46%	62%
R4	211	+	36,020–36,229	184	6
		+	130,730–130,940	96.84%	3%
R5	102	+	143,734–143,835	120	2
		+	233,534–233,634	98.36%	2%
R6	110	+	49,216–49,324	78	1
		+	205,946–206,055	98.73%	1.27%

## Data Availability

The raw sequencing data for the Illumina and Nanopore platforms and the mitochondrial genome sequences have been deposited in NCBI (https://www.ncbi.nlm.nih.gov/ (accessed on 6 March 2022)) with accession numbers: PRJNA782863, SAMN23402378, SRR17907850, SRR17908660, OK563725, and OK563726. The sample has been stored at the Huizhou University, Huizhou, China (voucher number: VR202101).

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
