# Peer review of "Sequence Analysis of the Complete Mitochondrial Genome of a Medicinal Plant, Vitex rotundifolia Linnaeus f. (Lamiales: Lamiaceae)"

_genes, 2022, doi:10.3390/genes13050839_

Round 1

Reviewer 1 Report

Dear Authors,

The current manuscript describes the mitochondrial DNA of a medicinal plant from China. In my honest opinion, the current version requires a deep English editing to quailify for publishing in this or any other journal. Moreover, authors should check species names along the text, in some instances not in italics. Some parts of the results should be explained in more detail. Thus I recommend this manuscript undergo major revisions.

Several citations are missing along the manuscript, such as Nextdenovo (L82), Getorganelle (Wrongly spelled at L86 as "Getarganelle"), as well as versions of some of the softwares employed (SAMtools, L89).

Some aspects of the results are quite confusing result section. EspecificalIy do not fully comprehend the intent nor what authors describe in section 3.2.

As I can understand from section 3.1, authors assembled a single MT DNA. Quote: "The mitochondrial genome of V. rotundifolia was assembled as one circular molecule of length", which I guess s the so called "master circle". Nevertheless, they proppose alternate versions of the mitochondrial DNA in figure 4 (please, state clearly what are colours and Rbs) and here things get quite messed. Do those alternate circles represent sub-genomes found as alternate shorter versions of the MT DNA within the assembly? Is this model solely a representation of the propossed evolution of these repeats? Please, correct this and reconstruct this section so that this is clearly stated.

Author Response

Response to Reviewer 1 Comments

We are grateful for your careful review and valuable comments on our manuscript. We have made the changes following the comments and please see our point-to-point responses.

Point 1: The current manuscript describes the mitochondrial DNA of a medicinal plant from China. In my honest opinion, the current version requires a deep English editing to quailify for publishing in this or any other journal.

Response 1: thanks for the suggestion. We have modified some language errors and our manuscript has undergone English language editing by MDPI.

Point 2: Moreover, authors should check species names along the text, in some instances not in italics.

Response 2: thanks for the careful checking. We have checked the “V. rotundifolia” and kept it in italics which appeared 62 times in the text.

Point 3: Some parts of the results should be explained in more detail. Thus I recommend this manuscript undergo major revisions.

Response 3: thanks for the suggestion. We have modified the result section 3.2 and explained our findings in more detail.

Point 4: Several citations are missing along the manuscript, such as Nextdenovo (L82), Getorganelle (Wrongly spelled at L86 as "Getarganelle"), as well as versions of some of the softwares employed (SAMtools, L89).

Response 4: thanks for the careful checking. We have added the citations of the Nextdenovo and GetOrganelle, and the versions of BWA and SAMtools in the manuscript.

Point 5: Some aspects of the results are quite confusing result section. EspecificalIy do not fully comprehend the intent nor what authors describe in section 3.2.

Response 5: as a suggestion. We have modified the result in section 3.2 and please see our new version of the manuscript.

Point 6: As I can understand from section 3.1, authors assembled a single MT DNA. Quote: "The mitochondrial genome of V. rotundifolia was assembled as one circular molecule of length", which I guess s the so called "master circle". Nevertheless, they proppose alternate versions of the mitochondrial DNA in figure 4 (please, state clearly what are colours and Rbs) and here things get quite messed. Do those alternate circles represent sub-genomes found as alternate shorter versions of the MT DNA within the assembly? Is this model solely a representation of the propossed evolution of these repeats? Please, correct this and reconstruct this section so that this is clearly stated.

Response 6:

6.1.We have modified the legend of Figure 4 and stated clearly what are colors and Rbs. Please see our new version of the manuscript.

6.2. Yes, those alternate circles represent sub-genomes found as alternate shorter versions of the MT DNA within the assembly.

6.3. This model has noted solely a representation of the proposed evolution of these repeats

As suggestion. We have added the sentences “These alternate circles represent sub-genomes found as alternate shorter versions of the MT DNA within our assembly of the V. rotundifolia mitochondrial genome.” and “It should be taken into notes that this model was not the sole representation of the proposed evolution of these repeats.” in the result section 3.2.

Reviewer 2 Report

Manuscript has quality work but need proper editing by native English Speaker.

Pls see highlighted part in manuscript attached file.

Author Response

Response to Reviewer 2 Comments

We are grateful for your careful review and valuable comments on our manuscript. We have made the changes following the comments and please see our point-to-point responses.

Point 1: Manuscript has quality work but need proper editing by native English Speaker.

Response 1: thanks for the suggestion. We have modified some language errors and our manuscript has undergone English language editing by MDPI.

Point 2: Pls see highlighted part in manuscript attached file.

Response 2: thanks for the careful checking. We have modified 12 parts which were highlighted in the manuscript attached file and please see our new version of the manuscript.